# Metabolic Hormones Modulate Macrophage Inflammatory Responses

**DOI:** 10.3390/cancers13184661

**Published:** 2021-09-17

**Authors:** Matthew J. Batty, Gwladys Chabrier, Alanah Sheridan, Matthew C. Gage

**Affiliations:** Department of Comparative Biomedical Sciences, Royal Veterinary College, 4 Royal College Street, London NW1 0TU, UK; mb2439@cam.ac.uk (M.J.B.); gchabrier@rvc.ac.uk (G.C.); asheridan20@rvc.ac.uk (A.S.)

**Keywords:** macrophages, hormones, TAMs, obesity, polarisation, metabolism, inflammation, cancer

## Abstract

**Simple Summary:**

Macrophages are a type of immune cell which play an important role in the development of cancer. Obesity increases the risk of cancer and obesity also causes disruption to the normal levels of hormones that are produced to coordinate metabolism. Recent research now shows that these metabolic hormones also play important roles in macrophage immune responses and so through macrophages, disrupted metabolic hormone levels may promote cancer. This review article aims to highlight and summarise these recent findings so that the scientific community may better understand how important this new area of research is, and how these findings can be capitalised on for future scientific studies.

**Abstract:**

Macrophages are phagocytotic leukocytes that play an important role in the innate immune response and have established roles in metabolic diseases and cancer progression. Increased adiposity in obese individuals leads to dysregulation of many hormones including those whose functions are to coordinate metabolism. Recent evidence suggests additional roles of these metabolic hormones in modulating macrophage inflammatory responses. In this review, we highlight key metabolic hormones and summarise their influence on the inflammatory response of macrophages and consider how, in turn, these hormones may influence the development of different cancer types through the modulation of macrophage functions.

## 1. Metabolic Hormones Modulate Macrophage Inflammatory Responses

### 1.1. Introduction

Hormones are ubiquitous chemical messengers that mediate physiological communication. Classically, they are defined as being produced by specialised cells within endocrine glands and released into the bloodstream in which they are carried until they reach their target cells. A tightly controlled spatiotemporal network of hormone signals mediates crosstalk within and between different organ systems to maintain healthy homeostasis. Disturbances to this network as a result of diet, lifestyle, or environmental factors can lead to obesity and diseases such as cancer [1]. Increased adiposity in obese individuals leads to dysregulation of many hormones [2] including those whose functions are to coordinate metabolism (which we refer to as ‘metabolic hormones’) (Table 1 and Figure 1A). Macrophages are phagocytotic leukocytes that play an important role in inflammation and have established roles in metabolic diseases [3] and cancer progression [4,5]. Recent evidence suggests many metabolic hormones play additional roles in inflammation, which includes modulating macrophage inflammatory responses.

This review highlights key classical metabolic hormones (Table 1 and Figure 1A) that become dysregulated in obesity and cancer and discusses their emerging roles in macrophage inflammatory responses (Figure 1B)—which have the potential to influence cancer progression (Figure 2). In this review, we have also included the sex hormones estrogen and testosterone due to their important direct and indirect roles in metabolism through their influence on body fat distribution and effect on cancer sexual dimorphism (Table 1 and Figure 1A).

### 1.2. The Role of Macrophages in Cancer Development

Macrophages are phagocytic cells of the hematopoietic lineage that play a central role in the innate immune response and also have established roles in metabolic diseases [3] and cancer initiation, malignancy, and metastasis [4,5]. Macrophages pervade almost every organ system and can exhibit a wide range of phenotypes depending on their particular microenvironment. During the last two decades, the conceptual framework for macrophage activation has evolved. Initially, macrophages were polarized into classically (M1) or alternatively (M2) activated cells [6] representing two polar inflammatory or anti-inflammatory extremes, respectively. The M1 vs. M2 model has been useful in describing immune responses during acute infections, allergies, asthma, and obesity [7]. However, observations from macrophages involved in chronic inflammation such as type 2 diabetes and cancer strongly suggest a much broader, context-dependent transcriptional repertoire in which macrophages adopt a spectrum of phenotypes that go beyond the rigid M1/M2 nomenclature [6,8]. Recent transcriptomics studies have now made considerable contributions to a better understanding of immune cell function and regulation; there are now at least nine distinct macrophage activation programs recognised [8], and within these programs, there are multiple unique gene expression signatures that enable macrophages to exist in a spectrum of activation states [8]. Nevertheless, to better compare the findings of the literature referenced within this review and in the absence of a framework to more accurately reflect the new macrophage phenotype spectrum, we will continue to use the M1/M2 convention as appropriate.

Tissue-resident macrophages may originate from yolk sac-derived erythromyeloid progenitors or circulating monocytes from bone marrow resident haematopoietic stem cells [9]. Tissue-resident macrophages display specific characteristics local to the tissues they reside in [10], which influence their function and hence their effect on their surrounding tissues. Macrophages can be highly influential on tumour development through the induction of inflammation, stimulation of neoangiogenesis, immune suppression or induction of metastasis. Macrophages that populate a tumour’s surrounding environment (the tumour microenvironment (TME)) are referred to as tumour-associated macrophages (TAMs); we direct the reader to the very recent review by Cendrowicz et al. [11] for a detailed description of the contribution of TAMs to the formation and development of tumours. Both clinical and experimental evidence has found that a high density of TAMs within a TME is strongly correlated with poor prognosis and reduced survival in a number of cancer types [12]. While TAMs have been found to display a wide spectrum of phenotypes, the majority are reported to have M2-like immunosuppressive properties due to the higher expression of IL-10 and TGF-β in TMEs, which is thought to help tumours evade cancer cell elimination by the immune system [12]. In contrast, pro-inflammatory M1-like TAMs are thought to establish a tumour-inhibiting phenotype by allowing tumoricidal activity to resume through the reversal of immunosuppressive mechanisms. Macrophages express hormone receptors [13] and because of the systemic nature of these metabolic hormones and the significant role that macrophages play in tumour development, the potential of dysregulated hormone levels to modulate tumour microenvironments and hence macrophage inflammatory responses may be significant.

## 2. Metabolic Hormones Play Roles in Modulating Macrophage Inflammatory Responses

### 2.1. Cholecystokinin (CCK)

#### 2.1.1. Origin and Function

The peptide hormone Cholecystokinin (CCK) is well-established as being a metabolic hormone secreted from I cells of the small intestine when high levels of dietary fatty acids or proteins are detected [14]. CCK regulates digestion by stimulating the release of digestive enzymes and insulin from the pancreas and mediates satiety by binding to CCK receptors in the vagal afferent neurons of the gut–brain axis [15]. However, CCK is also a neurotransmitter, growth factor and anti-inflammatory cytokine expressed as multiple different bioactive peptides by neurons, endocrine, and epithelial cells (recently reviewed in [16]). The effects of CCK are mediated through two types of receptors; CCK1R and CCK2R [17,18]. CCK1R is mainly located in peripheral tissues and shows higher selectivity for CCK than CCK2R [19].

#### 2.1.2. CCK and Cancer Association

Diets rich in long-chain saturated fatty acids lead to the overexpression of CCK which alongside obesity, is a significant risk factor for pancreatic cancer [20,21] and elevated CCK levels are also associated with the development of pancreatic metastases in mice [22]. CCK receptors can be over-expressed in a range of human cancers including stomach, pancreas, colon, rectum, oesophagus, lung, and liver [23,24].

#### 2.1.3. CCK Modulates Macrophage Inflammatory Responses

Studies to date indicate an anti-inflammatory role of CCK in several diseases and animal models of disease [25,26,27,28,29,30,31,32] demonstrated by ablation of CCK or CCKR, or treatment with CCKR antagonists which exert pro-inflammatory effects [27,30,33]. Both CCK receptors are expressed in macrophages [34,35] although CCK-1R is the predominant mediator of CCK’s immunomodulatory effects [36]. The CCK-8 isoform negatively modulates macrophage functions such as phagocytosis and tissue infiltration [26,27,37] and inhibits inflammatory response through downregulation of CD68, ICAM-1, TGF-β, and TNFα gene expression and inhibition of NF-κB activity [30]. In peritoneal and pulmonary interstitial macrophages, CCK-8 treatment blocks LPS-induced IL-1β production, reduces nitric oxide production and attenuates iNOS and TNFα mRNA expression [29,36,38]. These studies indicate the mechanism through which CCK exerts anti-inflammatory effects is through modulation of p38 and NF-κB activity via inhibition of PKC and activation of the cAMP-PKA pathway [29,30,36,37,38]. It is therefore possible that the pancreatic tumour growth associated with CCK expression can be attributed to the capacity of CCK to promote macrophages to adopt a pro-tumour, M2 phenotype. 

### 2.2. FABP4

#### 2.2.1. Origin and Function

Fatty acid-binding proteins (FABP) are at least a nine-member family of 14–15 kDa proteins that facilitate the absorption and utilisation of water-insoluble dietary long-chain fatty acids [39]. The different family members are uniquely expressed in distinct tissues involved in active lipid metabolism, including adipocyte FABP (known as FABP4). FABP4 is highly expressed by mature adipocytes [40,41] and macrophages [42,43] and the major regulator of FABP4 signalling is peroxisome proliferator-activated receptor (PPAR) γ [43,44]. 

#### 2.2.2. FABP4 and Cancer Association

Evidence suggests that FABP4 levels impact diseases ranging from metabolic syndrome, type two diabetes, atherosclerosis [45,46,47,48] and various forms of cancer including breast, liver, colon, and ovarian [49,50,51,52]. FABP4 has been shown to promote tumour progression via enhancement of new blood vessel formation and tumour growth mediated by its effects on adipocytes and tumour cells [53,54]. However, in contrast to these findings, decreased levels of FABP4 have also been associated with hepatocellular carcinoma tumours and FABP4 was shown to suppress proliferation and invasion of hepatocellular carcinoma cells [50], suggesting that the influence of FABP4 on cancer development and progression may depend on the cancer type and microenvironment situation.

#### 2.2.3. FABP4 Modulates Macrophage Inflammatory Responses

FAPB4 is expressed in macrophages [42,43], with substantial crosstalk between macrophages and adipocytes occurring upon inflammatory activation [55]. The regulation of FABP4 signalling by PPARγ was demonstrated by Garin-Shkolnik et al. through FABP4 triggering proteasomal degradation of PPARγ which inhibited PPARγ-related functions [56] including its role in inhibiting the expression of inflammatory cytokines and directing the differentiation of immune cells towards anti-inflammatory phenotypes [57,58]. As PPARγ interferes with NF-κB, AP-1 and STAT transcriptional activity [58,59], it inhibits the upregulation of pro-inflammatory genes, such as IL-1β, IL-6, and TNFα. The repression of PPARγ is therefore associated with the initiation of inflammatory pathways and impaired alternative M2 macrophage activation [60]. Indeed, FABP4-deficient macrophages are seen to have reduced basal and stimulated expression of pro-inflammatory cytokines including TNFα, IL-1β, MCP-1, and IL-6 due to decreased/NF-κB activity and deficient activation-induced expression of iNOS [42,61,62]. Additionally, FABP4 is shown to exacerbate LPS-induced inflammation by forming a positive feedback loop with the JNK signalling cascade [63], and subsequently influences the production of inflammatory cytokines. However, in contrast, some studies have suggested that FABP4 may enhance the activities of PPARγ during the differentiation of macrophages, providing a positive feedback loop between the two proteins [64]. These-conflicting findings have been suggested to arise from FABP4 exerting a concentration dependent effect on PPARγ regulation. Other regulators that have been seen to induce FABP4 expression in macrophages include rapamycin [65], which can increase the expression of genes involved in cholesterol transport and triglyceride synthesis. Notably, intracellular FABP4 has been observed to enhance pro-tumour macrophage function. FABP4 is highly expressed in a small subset of TAMs of the CD11b + F4/80 + MHCII − Ly6C − CD11c − phenotype [49]. FABP4-positive TAMs accumulate in late-stage mammary tumours, promoting their growth through the enhancing effect FABP4 has on NF-κB expression, thereby increasing the secretion of pro-tumour IL-6 signalling. Indeed, genetic ablation or chemical inhibition of FABP4 in TAMs has been shown to suppress mammary tumour growth [49].

### 2.3. Gastrin

#### 2.3.1. Origin and Function

Gastrin is a stomach acid secretion-regulating peptide hormone produced by endocrine G-/gastrin cells in the pyloric antrum of the stomach, duodenum, and pancreas. The gastrin gene in humans encodes a 101 amino-acid precursor peptide, which is subsequently cleaved to generate progastrin before being cleaved once again to form gastrin itself—the dominant forms of which in human plasma are gastrin-34 and gastrin-17 [66]. Once synthesised, gastrin peptides are stored in the basal part of the G-cells until they are released either in response to food intake or induced by the neurotransmitter gastrin-releasing peptide (GRP) acting on basolateral receptors in the G cells. Once released, gastrin modulates its effects through the CCK receptors; CCK1R and CCK2R, the latter of which has a higher affinity for gastrin. 

#### 2.3.2. Gastrin and Cancer Association

Gastrin has been observed to directly induce the expression of pro-inflammatory molecules such as IL-8, CINC-1 and the enzyme COX-2 in gastric epithelial cells [67,68]. COX enzymes are known to catalyse the synthesis of prostaglandins, a pathway shown to play an important role in cancers. Furthermore, COX-2 inhibition has been shown to suppress cell proliferation and induce apoptosis in various gastrointestinal cancer cell lines in vitro [69]. The pro-inflammatory effect of gastrin has also been hypothesised to play a role in cancer initiation through its association with *H. pylori* which is known to induce gastric cancer development and progression [70]. Gastrin-releasing peptide (GRP) and its receptor (GRPR) have also been linked to cancerous malignancies [71] and GRPR has been shown to induce the release of IL-8 and vascular growth factor in the case in human prostate cancer cell lines [72].

#### 2.3.3. Gastrin Modulates Macrophage Inflammatory Responses

Both CCK1R and CCK2R have been identified in an array of human leukocyte cell types, including lamina propria macrophages [73], peripheral blood mononucleocytes [34,74], circulating polymorphonuclear leukocytes (PMNs) [75] and also PMNs found within human malignant colorectal tumours [76]. One of the first links between gastrin and macrophages was provided by Okahata et al. in 1985 when they demonstrated that gastrin treatment resulted in increased immunoreactivity in pure populations of human PMNs [77] and further in vitro studies have shown that murine macrophages and human PMNs treated with gastrin induces chemotaxis and increases adherence and phagocytosis [78,79]. Alvarez et al. also noted that gastrin treatment increased leukocyte rolling and adhesion, decreased rolling velocity and increased leukocyte infiltration into the interstitium and as CCK2R, but not CCK1R antagonists abrogated these effects it is thought that CCK2R mediates gastrin’s inflammatory effects [80]. The gastrin receptor CCK2R has been further implicated in the pro-inflammatory response due to its promoter containing an IFN-γ regulatory site [81]. In addition to these direct effects on macrophages, gastrin may modulate their function by acting on the surrounding tissue. Gastrin, mediated by CCK2R, is reported to induce the release of IL-8 from human endothelial cells and increase the synthesis of the adhesion molecules VCAM-1 and P-selectin, resulting in increased adhesivity for human mononuclear leukocytes [82,83]. 

### 2.4. Ghrelin 

#### 2.4.1. Origin and Function

Ghrelin is an orexigenic hormone primarily produced by enteroendocrine cells in the stomach. Although its highest levels are found in the stomach and intestine [84], it is also expressed in the brain [85] and by macrophages [86]. Ghrelin has been demonstrated to regulate food intake, energy expenditure, glucose homeostasis, adiposity, body weight, inflammation, and growth hormone (GH) secretion [87]. There are two isoforms of ghrelin as a result of post-translational modifications; desacyl-ghrelin and acyl-ghrelin. The desacetylated form of ghrelin, desacyl-ghrelin, can undergo octanoylation performed by ghrelin O-acyltransferase (GOAT) to become acyl-ghrelin, which is also referred to as the active although least abundant form of the hormone. Ghrelin’s function is mediated by the Growth Hormone Secretagogue Receptors (GHSR). The acyl-ghrelin form can bind to GHSR1α and GHSR1β forms [88,89,90]. These receptors are highly expressed in the brain, poorly expressed in adipose tissue, and are localized within several immune cell types including monocytes and macrophages [86,90,91,92]. Ghrelin is down-regulated in obese patients and up-regulated under conditions of negative energy balance in humans and mice [93,94].

#### 2.4.2. Ghrelin and Cancer Association

Ghrelin regulates several processes related to cancer progression recently reviewed in [95] including cellular proliferation, inflammation, and energy homeostasis. However, the precise relationship between the ghrelin axis and cancer development remains unclear and controversial due to conflicting results observed between in vitro [96] studies involving a range of tumour cells lines and clinical studies [97]. Several recent attempts have been made recently to standardise these findings [95,98] leading to the hypothesis that the systemic nature of ghrelin signalling may contribute to confounding local factors that make delineating ghrelin’s direct role in cancer progression complex. Furthermore, experimental variations such as different cell lines and dosages used, add additional difficulties to interpreting the effects of ghrelin from these in vitro studies. In vivo animal and human studies have also revealed that ghrelin expression is often downregulated in cancer tissues and blood plasma while having no clear correlation with tumour development [99]. 

#### 2.4.3. Ghrelin Modulates Macrophage Inflammatory Responses

In macrophages the effects of ghrelin are complex; exogenous ghrelin treatment in the RAW264.7 macrophage cell line has been shown to inhibit LPS-induced production of pro-inflammatory cytokines IL-1β, TNFα and promote the release of the anti-inflammatory marker IL-10 [86]. However, in contrast, acyl ghrelin treatment in RAW264.7 macrophages promoted macrophage polarization to M1 under an inflammatory state by enhancing the effect of LPS and weakening the effect of IL-4 [100]. Although both studies used the same in vitro model, the doses of LPS used and the incubation time differed between the two studies (1000 ng/mL vs. 10 ng/mL) which may explain the contradictory results. In vivo work [101] has demonstrated that GHSR knockout, in aged mice, promotes macrophage phenotypic shift towards an anti-inflammatory M2 state. Yet, in contradiction to these results, a separate study showed GHSR knockout in mice fed a high-fat diet, to have a beneficial effect on adipose tissue [100]. The mice had reduced adipose tissue inflammation, decreased macrophage infiltration, and improved insulin sensitivity. Moreover, macrophages were polarized into an M2 type. Several additional studies show ghrelin to be anti-inflammatory [102,103,104,105], thought to be the result of JNK inhibition and activation of the Wnt/β-catenin pathway [92,106,107]. Recently [108,109], the effects of ghrelin, and ghrelin analogue hexarelin, on macrophages induced with oxidised-LDL (ox-LDL) and observed inhibition of LOX1 gene expression as a result of reduced ox-LDL uptake and decreased TNFα and IL-6 release. Similar protective effects of ghrelin have been demonstrated in alveolar macrophages induced with LPS where ghrelin treatment was able to reduce the level of IL-1β, TNFα, and IL-6, as well as decreasing iNOS and Akt activity. LOX-1-NF-κB and NF-κB/iNOS pathways or Akt signalling are the pathways suggested to mediate these anti-inflammatory effects [108,109,110]. 

### 2.5. Incretins 

#### 2.5.1. Origin and Function

The incretin hormones; glucose-dependent insulinotropic polypeptide (GIP, also known as gastric inhibitory polypeptide) and glucagon-like polypeptide-1 (GLP-1) are secreted by the small bowel in response to nutrients and stimulate the release of insulin from pancreatic β cells. Both GIP and GLP-1 are rapidly degraded by dipeptidyl peptidase 4. GIP is a 42-amino acid peptide secreted by enteroendocrine K cells of the small intestine. Its action is mediated by gastric inhibitory peptide receptor (GIPR), which is expressed mainly in the pancreas but can also be found in other tissues and immune cells [111,112]. GLP-1 is expressed in L cells of the small and large intestines [113] and is derived from the proglucagon gene [114]. GLP-1 action is mediated by the glucagon-like polypeptide-1 receptor GLP1R [115]. 

#### 2.5.2. Incretins and Cancer Association

GLP1R is reportedly overexpressed in many tumour cell types [116]. However, the role of GLP1R overexpression in neoplasia remains unknown Incretin mimetic drugs such as exenatide and liraglutide are powerful and widely used treatments for type II diabetes [117]. Like GLP-1 and GIP, they bind to GLP1R on the pancreatic β cells to stimulate insulin secretion. Exenatide has been shown to promote apoptosis in SKOV-3 and CAOV-3 human ovarian cancer cell lines and reduce the production of the pro-metastatic adhesion molecules ICAM-1 and VCAM-1 by TNFα stimulated vascular endothelial cells [118]. Similar observations have been made in studies using breast [119], colon [120], and ovarian [121] cancer cell lines, likely due to the inhibition of the PI3K/Akt signaling pathway by exenatide. However, whilst concerns have been raised over the safety of incretin-based treatments for those living with both cancer and diabetes meta-analysis of clinical data has not revealed any association between the development of lung [122], gastrointestinal [123] or pancreatic [124] cancer and the prescriptions of the incretin analogues.

#### 2.5.3. Incretins Modulate Macrophage Inflammatory Responses

Incretins have demonstrated overall metabolic benefits and anti-inflammatory effects in macrophages in the context of inflammation-mediated obesity [125]. Treatment with GLP1, or GLP1 agonist, improves insulin sensitivity, glucose, and insulin tolerance, normalises glycemia, and reduces fat mass, adipocyte size and macrophage number in adipose tissue as a result of macrophage infiltration inhibition [126,127,128]. In vitro studies on several different models of macrophage cells (peritoneal macrophages, LPS treated RAW264.7 cells and adipose tissue macrophages (ATM)) have consistently shown the anti-inflammatory effect of GLP1 and its agonists. These studies have shown a decrease in pro-inflammatory markers (iNOS, IL-1β, IL-6, TNFα and MCP1, MMP2, MMP9, ROS) [129,130,131]) while increasing anti-inflammatory markers (IL-10, mannose receptor-1 (MRC-1), macrophage galectin-1 (MGL-1), arginine-1 (ARG-1)) [128,129,130,132], PGE2 and COX2 mRNA and COX2 protein level [129]. Studies have demonstrated these effects to be mediated through JNK inhibition, STAT3 activation [132,133], and also the cAMP/PKA/NF-κB signalling pathway [126,128,129,130,132]. In vitro, GLP1 has been shown to decrease cell migration through its reduction of MCP-1 and MMP-9 activity which may be mediated by the inhibition of Nf-κB, JNK, ERK, and p38 in LPS activated macrophages [134,135]. GLP1 and GIP inhibitor DPP4 (also known as CD26) enhances the expression of TNFα and IL-6 mRNA and protein in THP-1 cells [136] while GIP prevents proinflammatory macrophage activation leading to reduced LPS-induced IL-6 secretion [134]. 

### 2.6. Insulin

#### 2.6.1. Origin and Function

Insulin is an anabolic hormone secreted by the β cells of the pancreas. Insulin regulates glucose homeostasis, lipid metabolism [137] and cell growth [138]. The effects of insulin are mediated through the insulin receptor (INSR) [139,140] and the insulin-like growth factor 1 receptor (IGF-1R) [141]. Upon binding to its receptor, insulin activates two major downstream pathways, the PI3K pathway which mediates its metabolic effects [142] including the translocation of GLUT4 in metabolic tissues such as muscle and adipose, and the MAPK kinase pathway [143] which regulates mitogenesis and growth. The insulin receptor has also been shown to act directly as a transcription factor [144], which may illuminate previously unrecognised mechanisms of the long-term effects of insulin normal physiology and disease.

#### 2.6.2. Insulin and Cancer Association

Metabolic diseases such as obesity and type 2 diabetes that are characterised by hyperinsulinemia are associated with the development of several types of cancer including colon, breast, endometrium, oesophagus, kidney, liver, and pancreas [145,146,147,148]. Insulin can promote cancer through its mitogenic actions [149] and the INSR is often overexpressed in tumour cells [150,151,152]. More recently, a new mechanism has been demonstrated showing that hyperinsulinemia promotes epithelial tumourigenesis by abrogating cell competition [153].

#### 2.6.3. Insulin Modulates Macrophage Inflammatory Responses

Macrophages express insulin receptors [154] and intracellular signalling machinery. Existing reports demonstrate diverse effects that include promoting pro-inflammatory responses through increasing phagocytosis [155], and TNFα production [156]. However, other studies report an anti-inflammatory response involving decreased apoptosis [157] and decreased pro-inflammatory cytokine production [158,159]. Discrepancies may be due to inconsistencies in the macrophage cells used (mouse [160] or human cell lines [156,157]; mouse tissue macrophages [161] or human peripheral cells [162]), and lack of consistency of insulin concentration and duration used. We would direct the reader to the very recent review [163] detailing insulin’s inflammatory and anti-inflammatory effects on macrophages.

### 2.7. Insulin-Like Growth Factor-1 

#### 2.7.1. Origin and Function

Insulin-Like growth factor-1 (IGF-1) is a polypeptide hormone mainly produced in the liver where growth hormone (GH) binds to its receptor to drive most of the IGF-1 synthesis [164]. IGF1R is also expressed in adipocytes [165] endothelial cells and macrophages [166,167]. Its actions can be endocrine, paracrine, or autocrine and are mediated by the tyrosine kinase receptor IGF-1R. IGF-1 is involved in growth, regulation of metabolism, and inflammation [168]. Structural similarities with insulin enable IGF-1 to bind to both IGF-1 and insulin receptors [169]. IGF-1 signal transduction requires insulin receptor substrates (IRS) [170] and involves the downstream phosphoinositide-3 kinase (PI3-K) and MAPK activation pathways [171]. 

#### 2.7.2. IGF-1 and Cancer Association

Above normal levels of circulating IGF-1 are associated with an elevated risk of developing several primary cancer types including colorectal [172] and breast [173]. IGF-1 has been demonstrated to enable tumour growth by preventing apoptosis through the induction of the PI3K and MAPK signalling cascades [174]. Furthermore, IGF-1 has been shown to increase the migratory and proliferative capabilities of IGF-1R-overexpressing HCT116 colon cancer cells, leading to metastases in mice transfected with these cells [175]. Finally, IGF-1R is known to be overexpressed by cancer cells contributing further to the malignant phenotype [176].

#### 2.7.3. IGF-1 Modulates Macrophage Inflammatory Responses

Macrophages polarized into an anti-inflammatory M2 subtype express high levels of IGF-1 [166,177,178]. A recent study assessing IGF-1 implication in the development of obesity [166] using mice with myeloid cell-specific ablation of IGF-1R and challenged with a high-fat diet showed increased macrophage infiltration in adipose tissue leading to insulin resistance and also suggested an anti-inflammatory role for IGF-1 on macrophages. In contrast, however, other studies have reported IGF-1 to have a pro-inflammatory effect on macrophages; IGF-1 treatment on murine macrophages has been shown to increase production and expression of TNFα mediated by IGF-1R and tyrosine kinase factor activation [179], and to stimulate LDL uptake and cholesterol esterification [180]. Ablation of IGF-1R in macrophages was also seen to significantly decrease NLRP3 inflammasome-dependent caspase-1 and IL-1β activation when induced by ageing-relevant damage-associated molecular patterns (DAMPs) [181]. 

### 2.8. Leptin 

#### 2.8.1. Origin and Function

Leptin is a pleiotropic peptide hormone encoded by the *ob* gene and secreted by adipose tissue in proportion to its abundance [182]. It was first reported to control food intake and body weight through anorexigenic effects in the brain [183,184]. Leptin receptors (known as LEPR or OBR) are class I cytokine receptors expressed in many cell types, including macrophages [185] where Ob-Rb is the most well-characterized isoform [186,187].

#### 2.8.2. Leptin and Cancer Association

The role of the leptin-Ob-Rb signalling axis in cancer development is well documented with both leptin and OBR becoming overexpressed in several cancer types including head and neck, pancreatic, and breast cancer [188]. Leptin signalling has been implicated as a driver of angiogenesis in colorectal cancer [189] and glioblastoma [190], resulting in metastasis and tumour growth. Leptin increases the expression of inflammatory cytokines such as TNFα and IL-1β which contributes to tumour-associated inflammation and subsequently, immunosuppression of tumouricidal CD8+ cytotoxic T cells in breast cancer [191]. Furthermore, leptin is known to have direct effects on ovarian cancer cell proliferation and growth by activating the PI3K/Akt and MEK/ERK1/2 signalling pathways [192]. Obesity is also associated with elevated serum leptin levels, with many obese individuals experiencing ‘leptin resistance’ in which leptin is no longer able to effectively regulate food intake. The exact reason for the upregulation of leptin during obesity is unknown, but it has been speculated that leptin may be a critical link between obesity and cancer [193,194].

#### 2.8.3. Leptin Modulates Macrophage Inflammatory Responses

Leptin is an adipokine, a member of the superfamily of cytokines [195] and is implicated in inflammation, infection, and immune responses [182]. Genetic abnormalities in leptin or leptin receptors are reported to impair macrophage phagocytosis and promote pro-inflammatory cytokines production, while leptin treatment increases both [196]. Similarly, in vitro studies on the murine macrophage J774A.1 cell line, human monocyte-enriched mononuclear cells, LPS stimulated Kupffer cells and human adipose macrophages have shown that leptin can stimulate the phagocytotic activity of macrophages [197] and the proliferation and activation of monocytes [198]. In addition, leptin may stimulate production of proinflammatory cytokines TNFα, resistin, IL-6, and IL-1β, IL-1Ra, IL-10, MCP-1, and MIP-1α and enhance CC-chemokine ligand expression [199,200,201]. These effects might be mediated through activation of a JAK2-STAT3 pathway [199] and may also activate ERK1/2, P8 MAPK, JNK, AMPK, PKC and PI3K/Akt pathways [185,202,203,204,205]. 

### 2.9. Neuropeptide Y (NPY) and Peptide YY (PYY)

#### 2.9.1. Origin and Function

Both NPY and PYY belong to a family of neuropeptides bearing a close resemblance to each other, consisting of 36-amino acids with a unique hairpin turn called the PP-fold [206]. NPY is highly abundant and is found in all levels of the gut-brain axis as well as being highly expressed in the central nervous system where it is widely known for its activity as a regulator of food intake and energy balance [207]. PYY is almost exclusively associated with the digestive system and is predominantly expressed in L cells in the ileal and colonic mucosa and released into the bloodstream post-prandially in proportion to calorie intake [207,208,209,210]. NPY and PYY peptides can also be truncated, yielding the fragments NPY(3-36) and PYY(3-36) [211]. In humans, NPY and PYY’s functions are mediated by diverse G-protein coupled Y receptor subtypes, of which seven have been noted, but only four are widely functional (Y1, Y2, Y3 and Y4). NPY(1-36) and PYY(1-36) are thought to bind to all the receptors with an equal affinity, whilst NPY(3-36) and PYY(3-36) exhibit the highest affinity for Y2 [212,213,214]. 

#### 2.9.2. NPY and PYY and Cancer Association

Investigation of PYY and NPY, have collectively revealed that they are implicated in a variety of inflammatory disorders, such as autoimmune diseases, asthma, atherosclerosis, and cancer [215,216,217]. Y receptors have recently attracted attention due to their overexpression in various human cancers, including breast carcinomas and neuroblastomas, creating interest in their use as a possible target for cancer imaging and therapy [218]. The Y receptors mediate tumour development through their direct effect on cancer and endothelial cells promoting tumour cell proliferation, survival, and migration, as well as angiogenesis) [219]. 

#### 2.9.3. NPY and PYY Modulate Macrophage Inflammatory Responses

In macrophages, neuropeptides have been found to exert varying effects depending on the age of the subject. In one of the first studies examining their role in macrophage function, both NPY and PYY were found to increase adhesion, chemotaxis, and phagocytosis in murine peritoneal macrophages, as well as increasing the production of superoxide anions in young adult mice [220]. The authors noted that this effect was produced through the stimulation of PKC due to a significant increase in its activation following NPY and PYY treatment. However, in more aged mice, this effect was potentiated, with chemotaxis and phagocytosis being decreased. These changes have been hypothesised to be dependent on the activity of dipeptidyl peptidase 4, an enzyme that terminates the activity of neuropeptides on the Y1 receptor subtype and whose activity is seen to change with age [221]. This age-dependent impact in modulating the immune response was also found to be true concerning PYY and NPY acting via Y1 receptors to potentiate nitric oxide production in rat peritoneal macrophages, with production being suppressed in older rat cells [222]. Y receptors are known to be widely expressed in immune cells, particularly Y1, which has been found in almost every type of immune cell [223]. 

The expression of Y receptors is also significantly upregulated after antigen or inflammatory stimulation [224,225,226]. Studies have also demonstrated the ability of neuropeptides to modulate macrophage cytokine secretion. However, contradictory results have been found. Y1 ablation in macrophages has been seen to lead to an increased pro-inflammatory phenotype displaying increased inflammatory response and exacerbated secretion of MCP-1 and TNFα, and a similar response was seen in macrophages isolated from double NPY and PYY knockout mice, suggesting an anti-inflammatory role [227]. Additionally, NPY was shown to decrease the production of TNFα and IL-1β following LPS treatment [228,229] and increase that of TGFβ1 in RAW264.7 cells [230]. In contrast, other studies have reported NPY to increase the production of pro-inflammatory mediators, with NPY being found to significantly increase the expression of TNFα, C-reactive protein, MCP-1 and reactive oxygen species in RAW264.7 macrophages mediated by the Y1 receptor [231]. NPY has also been shown to stimulate IL-1β secretion in aged animal macrophages [232]. Furthermore, in whole blood cells from healthy subjects, NPY upregulated IL-6, IL-1β and TNFα production [233]. Some suggestions for the observed duality have been differences in species, cell type and cell environment. Additionally, the activation of different Y receptor types is seen to mediate different effects, and there is evidence that along with Y1, Y4 and 5 may also play a role in cytokine modulation [227]. A relatively recent study by Cheng et al. found sympathetic stimulation of prostate cancer cells in vitro led to the release of NPY, which in turn was seen to promote myeloid cell trafficking and increased IL6 synthesis in TAMs, promoting tumorigenesis [234]. However, the connections between neuropeptides, immune regulation and cancerous disease have not yet been fully explored, and indeed it may be found that neuropeptides have divergent effects on immune cells in cancer development as observed in their general effects on macrophage function. 

### 2.10. Estrogen

#### 2.10.1. Origin and Function

Estrogens are a class of steroid hormones and are the main female sex hormones, but also play an essential role in both male and female reproductive and non-reproductive processes [235,236]. Estrogen is predominantly synthesised in the gonads in pre-menopausal women; however, non-gonadal sites such as adipose tissue, bone, skin, and the liver and brain can also produce a small but significant amount of estrogen. In humans, estradiol (E2 or 17β-estradiol) is the most prevalent and active form of estrogen. Estrogens play an important role in body weight, fat distribution, energy expenditure and metabolism [237]. 

The effects of estrogen are mediated by two intracellular estrogen receptors (ERs), ERα and ERβ [238] and by a plasma membrane protein, G protein-coupled estrogen receptor (GPER) [239]. ERα is predominantly expressed in the uterus, ovaries, and breasts, while expression of ERβ is mainly found in the nervous system, ovaries, cardiovascular system, and the male reproductive system [238]; however, all of these receptors are expressed in both rodent and human macrophages [240,241,242,243,244]. 

#### 2.10.2. Estrogen and Cancer Association

Obesity is often associated with elevated estrogen levels [245] and estrogens are thought to be involved in the sex differences observed to cancer susceptibility [246] and survival rates, with men having higher risk and mortality than women across various cancer types, excluding notable exceptions such as breast cancer [247,248]. Various experimental studies have demonstrated ER activity to exert anti-cancer effects recently reviewed in [249]. One example of this is ERβ activation shown to suppress the viability and migration of PC-3 and DU145 prostate cancer cell lines by suppressing the inflammatory NF-κB signalling pathway [250]. Another study reported that high levels of ERα expression in cancer-associated fibroblasts suppressed prostate cancer invasion by reducing macrophage migration via its suppression of the chemokine CCL5 [251]. In contrast, other experimental studies implicate estrogen as a potential mediator of tumour immune evasion through its association with the accumulation and increased activity of myeloid-derived suppressor cells, a set of immune cells associated with tumours and treatment resistance [252]. This relationship is thought to arise through ERα mediated activation of the STAT3 pathway, which has separately been linked to cancer cell survival and the expansion of myeloid-derived suppressor cells in cancerous growths [253].

#### 2.10.3. Estrogen Modulates Macrophage Inflammatory Responses

Estrogen has been shown to alter macrophage function via its receptors in a variety of ways including their proliferation [254], polarisation and cytokine production [246]. However, it should be noted that RNA transcription levels in resting macrophages indicate that ERα and GPER are mainly responsible for mediating estrogen action under physiological conditions [255]. The exact splice variants remain controversial, with different variants being reported as the major receptor present in macrophages across various reproductive and non-reproductive tissues and between the sexes [240,244]. 

During the past decade, the ability of macrophages to proliferate locally has been demonstrated in adipose tissue during obesity [256] and to be an important driver of atherosclerosis development in advanced atherosclerotic plaques [257,258,259,260]. A comprehensive description of the genomic responses induced in peritoneal macrophages by a mimicked “estrogen surge” found that estrogen regulates several genes associated with proliferation [254]. However, the exact mechanism by which estrogen is exerting its effect on macrophage proliferation remains unclear. Although estrogen response elements appear to be present in the promoter region of several cell-cycle genes, including Chafa1a, CcnB2 and Wee1, suggesting estrogen may have a direct effect, in vitro assays studying estrogen’s effect on peritoneal cells were unable to support macrophage proliferation. It therefore cannot be ruled out that indirect mechanisms involving other peritoneal cells were responsible for the observed increase in proliferative genes observed. 

The study by Pepe et al. [254], also showed that macrophages were initially found to adopt an M2-resembling subtype upon exposure to the mimicked “estrogen surge” with conversion to the pro-resolving phenotype as shown by the induction of the key immunosuppressive cytokine IL-10. Similarly, Campbell et al. [261], demonstrated that treatment with E2 or an ERβ agonist significantly dampened the 6 h post-stimulation increase in Nos2 expression usually associated with pro-inflammatory activation of BMDMs upon LPS and IFN-γ stimulation [261], suggesting that estrogen induces an inhibitory effect on pro-inflammatory polarisation. In another study, pre-treatment with ERα agonist was seen to strongly induce Arg1 expression (a later marker of alternatively activated, M2-like macrophages) after only 6 h of stimulation with IL-4, suggesting a direct ERα-mediated transcriptional effect. However, in LPS-induced inflammatory conditions, E2 was once again found to influence macrophage polarisation resolution upon inflammatory insult by accelerating the progression of the inflammatory process towards the IL-10 dependent “acquired deactivation” phenotype via SOC3 and STAT3 signalling pathways [262]. In addition, opposing reports regarding estrogen’s role in polarisation have also been noted. Yang et al. [263] found estradiol to repress alternative activation in TAMs through the inhibition of the JAK1-STAT6 pathway via ERβ, thereby inhibiting hepatocellular carcinoma tumour growth [263]. In ovariectomised rats, estradiol was found to promote M1-like macrophages through cadherin-11 after the induction of temporomandibular joint inflammation [264]. This discrepancy regarding estrogen’s effect on macrophage polarisation state may arise from differences in the specific microenvironment and subsequent ER activation pathways, as a ‘yin-yang’ relationship has been demonstrated with respect to estrogens’ effect on tissues mediated by the activation state of ERα and ERβ. For example, in hormone-related cancers, ERα has been shown to promote proliferative effects, whereas ERβ is found to inhibit cancer cell proliferation [265]. 

Seemingly contradictory data also exists regarding estrogen-mediated effects on macrophage cytokine production. One of the ways estrogen’s apparent heterogeneous nature appears most evident is in its enhancement or suppression of TNFα, IL-6 and IL-1β gene expression [266,267,268,269,270,271,272,273]. However, this discordance has not been as readily explained by discrepancies in species, estrogen concentration or other culture conditions, leading to the hypothesis that there may be multiple distinct pathways by which estrogen influences cytokine expression. The enhancement or suppression of various cytokines may depend on the specific activation of these distinct pathways in the individual cell context. Alternatively, the increase or decrease in cytokine production seen may be due to certain intrinsic or extrinsic coregulators of estrogen action that are able to change the response, based again on the particular experimental conditions. This discrepancy, particularly with respect to in vivo studies, may also arise in part due to pharmacological differences between hormonal replacement and endogenous estrogen secretion. One example of this is a study in which estrogen’s effect on IL-6 concentration in healthy fertile rats did not correspond with that of ovariectomised rats given exogenous E2 [267]. Moreover, there is evidence that estrogen aids tumoral M2-like TAM invasion and promotes macrophage secretion of tumour growth factors, such as VEGF [274]. However, Yang et al. noted opposite findings of estrogen’s effect on TAM polarisation [263]. Further studies are therefore needed to fully understand the complex interaction of estrogen and TAMs in context-specific situations.

### 2.11. Testosterone

#### 2.11.1. Origin and Function 

Testosterone is one of four androgen hormones in humans, the others being dihydrotestosterone (DHT) (a metabolite of testosterone), androstenedione and dehydroepiandrosterone [275]. Although DHT is the most potent of these androgens, testosterone is the principal sexual steroid hormone in men, with the highest concentration in adult male serum [276]. In males, testosterone is primarily synthesized in the testes’ Leydig cells and has a characteristic four ring C18 steroid structure [275]. Testosterone is known to exert genomic effects through binding to intracellular androgen receptors (ARs), which are ligand-inducible nuclear transcription factors [277]. However, rapid physiological responses to testosterone have also been reported. As these occur too quickly to be explained by the classical AR genomic pathway, it is now generally accepted that androgens must also exert non-genomic effects, which are assumed to be mediated through unconventional receptors in the plasma membrane [278]. 

#### 2.11.2. Testosterone and Cancer Association

Obesity is frequently associated with low androgen levels in men [279], whilst women with central obesity have higher total and free testosterone levels than normal-weight women [280]. Cancer sex-disparity in incidence, aggressiveness and prognosis has long been observed and testosterone’s modulating effect on the immune system has been investigated in relation to cancer development and progression. In one study of induced thyroid cancer in male mice, gonadectomy led to an upregulation of tumour-suppressor genes, *Glipr1* and *Sfrp1* [281] suggesting that testosterone promotes thyroid cancer progression through the suppressed expression of these genes. This suppression of *Glipr1* is also thought to impact the immune response through its modulation of Ccl5 secretion, a chemokine that plays important roles in chemotaxis and activation of immune cells. The reduced Ccl5 secretion associated with *Glipr1* knockdown led to reduced tumour infiltration by inflammatory macrophage and CD8+ T cytotoxic T cells, thereby aiding tumour immunity, as infiltration by these immune cells is usually associated with reduced tumour growth. Androgen deprivation therapy is also associated with increased infiltration of macrophages and T lymphocytes in prostate cancer patients [282]. Thus, testosterone’s observed cancer-promoting effects may be the result of its immunosuppressive ability; however, further investigation is warranted.

#### 2.11.3. Testosterone Modulates Macrophage Inflammatory Responses

Testosterone and other androgens, such as DHT, are generally regarded as immunosuppressors [283]. In line with this, testosterone deficiency has been associated with several disease states involving inflammation, such as cardiovascular disease [284], and various metabolic disorders such as type 2 diabetes mellitus [285]. Furthermore, testosterone replacement therapy has been reported to reduce circulating inflammatory cytokines in hypogonadal men, whilst promoting the secretion of the anti-inflammatory cytokine IL-10 [286]. 

Specifically, regarding macrophages, several studies suggest that testosterone can modulate macrophage cytokine production and macrophage activity/function. In vitro investigations have shown that androgen treatment diminishes the production of the pro-inflammatory cytokines TNFα and IL-1β in both rodent and human macrophage cell lines [287,288], and in a rat model of experimental autoimmune orchitis, testosterone replacement was found to down-regulate TNFα, IL-6 and MCP-1 mRNA expression in the testis, whilst inhibiting macrophage recruitment (simultaneously increasing the number of immunosuppressive regulatory T-cells) [289]. There is also evidence of testosterone exerting anti-inflammatory effects through regulating macrophage production of reactive oxygen intermediates [290] and nitrites via the inhibition of iNOS [291]. Alongside this, another mechanism by which androgens are thought to regulate macrophage action and exert immunosuppressive effects is through the downregulation of Toll-like receptor 4 (TLR4) [292]. Rettew et al. demonstrated that in vitro testosterone treatment of RAW 264.7 murine macrophage-like cells significantly decreased TLR4 expression, and further evidence of this was seen when castrated animals expressed elevated prostate TLR4 expression compared to intact [292,293]. The activation of TLR4 triggers downstream intracellular signalling cascades, including extracellular signal-regulated kinase (ERK), which mediate the secretion of inflammatory cytokines [294]. Therefore, it has been suggested that the TLR4 pathway may represent a key aspect of the increased inflammation seen with testosterone deficiency, and emerging studies have supported this, showing that removal of endogenous testosterone results in elevated ERK activity [295].

The exact molecular pathway by which testosterone alters macrophages’ immune responsiveness has not yet been fully elucidated. ARs are expressed in human macrophages [296,297]; however, in Friedl et al.’s investigation [291] of testosterone’s effect on nitric oxide synthesis, they noted that their observed findings were unlikely to be AR-dependent as the concentrations used in their experiment were much higher than the dissociation constant of the AR c. They therefore suggested that testosterone inhibits iNOS promoter activity via receptor-independent means. Thus, it is likely that both genomic and non-genomic testosterone mediatory pathways are present in macrophages. Indeed, ARs are noted to be associated with infiltrating macrophages in prostate cancer (PCa) development, with Cioni et al. recently demonstrating that AR activity in macrophage-like cells stimulates TREM-1 signalling, thereby promoting PCa-derived cancer cell migration and invasion [298].

### 2.12. Future Perspectives

Metabolic hormones are dysregulated in obesity, and obesity is a significant risk factor for cancer development [1]. Metabolic hormones are systemic in their nature and so have the capability to reach many different cell types including a range of tissue-resident macrophages, tumour-associated macrophages and their precursors. As evidenced throughout this review, a growing number of studies have reported new roles of these hormones beyond that of their classical functions of coordinating metabolism. These studies demonstrate a variety of regulatory effects of metabolic hormones on macrophage activity, including polarisation, cytokine secretion, migration, and phagocytosis. Some of the reported effects conflict (Figure 1B), describing pro- or anti-inflammatory actions on macrophages for the same hormone. These studies clearly demonstrate the significant impact that metabolic hormones may play in modulating macrophage function and now pave the way for further research to establish the relevance of these inflammatory-modulating properties in the context of metabolic hormone levels in health and disease. 

The complexity of this field can be further appreciated when acknowledging that during chronic diseases such as obesity and cancer, multiple metabolic hormones may be dysregulated simultaneously or at different times during disease progression. These dysregulated hormones may activate multiple intracellular signalling pathways in macrophages, and some of these pathways are shared between hormones (Figure 3). 

Identifying the influence that each hormone contributes and/or the net effects of these dysregulated hormones on macrophage responses is the challenge now presented, and this too may be cancer-specific, dependent on the individual tumour microenvironment and associated tumour macrophage type.

Studies directly linking the immunomodulatory mechanisms of metabolic hormones on macrophages to cancer development are limited. However, macrophages pervade almost every organ system and so the potential of dysregulated hormone levels to modulate macrophage biology into pro-cancerous phenotypes may be significant. Therefore, studies are urgently needed to better understand the interplay between these metabolic hormones, immune cells such as macrophages and cancer development. Such research may help identify novel cancer treatment strategies which may focus for example on normalising levels of dysregulated metabolic hormones or targeting the hormone receptors or aberrantly activated intracellular signalling pathways.

## Figures and Tables

**Figure 1 cancers-13-04661-f001:**
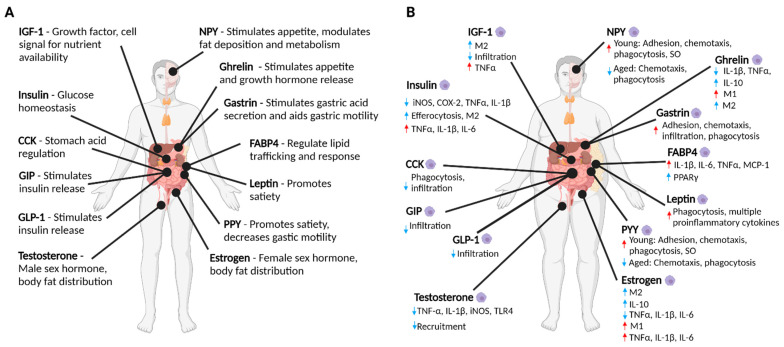
Schematic summarising (**A**) the classical functions of metabolic hormones in a healthy individual, and (**B**) how these hormones can modulate macrophage inflammatory responses when potentially dysregulated in an obese state. Red arrows indicate proinflammatory actions, blue arrows indicate anti-inflammatory actions. This figure was created with Biorender.com.

**Figure 2 cancers-13-04661-f002:**
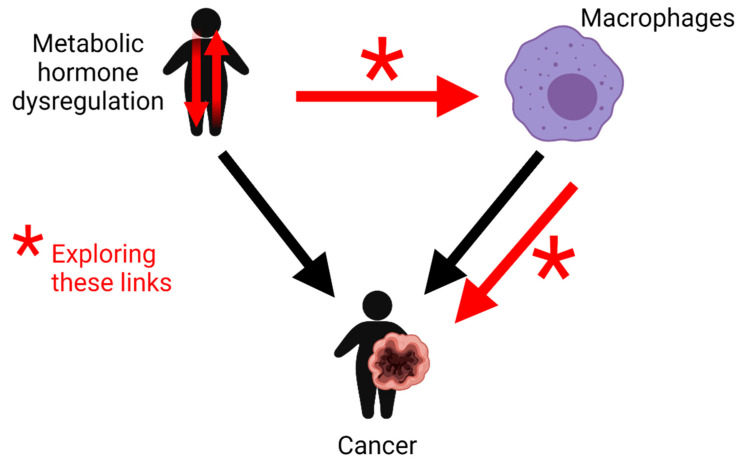
Schematic illustrating the links this review will explore (red arrows) between potential metabolic hormone dysregulation, macrophage inflammatory responses and cancer development. This figure was created with Biorender.com.

**Figure 3 cancers-13-04661-f003:**
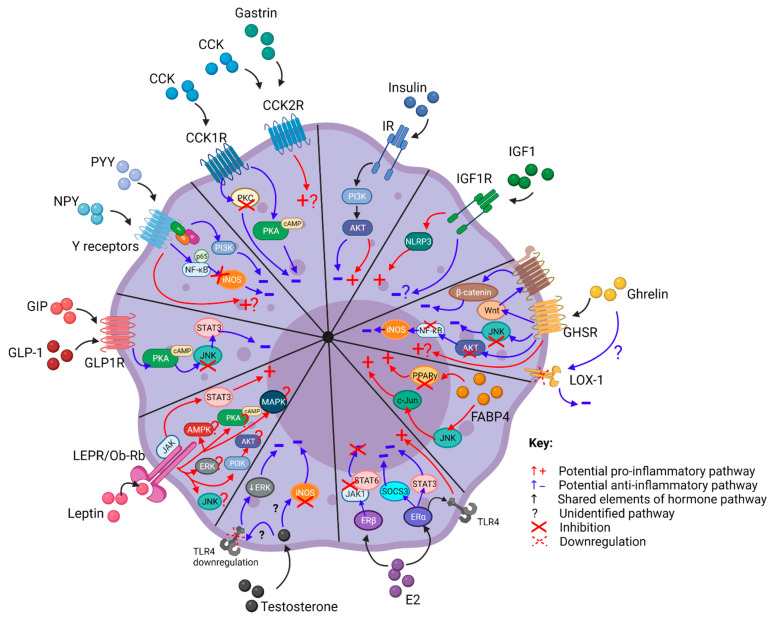
Schematic summarising the potential intracellular signalling mechanisms by which metabolic hormones affect macrophage inflammatory responses. This figure was created with Biorender.com.

**Table 1 cancers-13-04661-t001:** Summary of the metabolic hormones reviewed.

Hormone	Trigger	Origins	Metabolic Target	Receptor	Primary Metabolic Functions
CCK	Fatty acids, proteins	Small intestine I-cells	Pancreas	CCK1R & CCK2R	Stimulates release of digestive enzymes and insulin
FABP4	Lipolysis	Adipocytes Macrophages	Adipocytes Macrophages	PPARγ	Absorption of fatty acids M2 macrophage polarisation
Gastrin	Food intake Gastrin releasing peptide	Stomach G-cells Duodenum Pancreas	Stomach	CCK1R & CCK2R	Stomach acid regulation
Ghrelin	Food intake	Stomach Intestine Brain Macrophages	Brain Adipose tissue	GHSR	Regulates food intake, energy expenditure, glucose homeostasis, adiposity
GIP	Glucose, fatty acids	K-cells in the duodenum and jejunum	Pancreatic β-cells	GIPR	Stimulates insulin release
GLP-1	Hexose, fats	L-cells of the small intestine	Pancreatic β-cells Brain	GLP1R	Stimulates insulin release Induces satiety
Insulin	Hyperglycaemia	Pancreatic β-cells	Muscle Liver Adipose	INSR IGF1R	Glucose uptake Inhibition of gluconeogenesis
IGF-1	Growth Hormone (GH)	Liver Macrophages Adipocytes	Bones Smooth muscle Neurons	IGF1R INSR	Stimulates bone and tissue growth
Leptin	Food intake	Adipocytes	Brain	OBR	Regulation of food intake
NPY	Food intake (High levels of dietary fat and sugar)	Central nervous system	Central and peripheral nervous systems	NPY Receptors (GPCRs)	Regulation of food intake
PYY	Amino acids Short-chain fatty acids	L-cells of the ileum and colon	Central and peripheral nervous systems	PYY Receptors (GPCRs)	Gastric emptying Gut motility
Estrogen	Luteinizing hormone (LH)	Gonads, adipose tissue, bone, skin, liver & brain	Systemic	ERα Erβ GPER	Primary female sex hormone Fat distribution Metabolism
Testosterone	Luteinizing hormone (LH)	Leydig cells of the testis & adrenal glands	Systemic	AR	Primary male sex hormone Fat distribution Muscle mass

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
