# Peer review of "Metabolic Hormones Modulate Macrophage Inflammatory Responses"

_cancers, 2021, doi:10.3390/cancers13184661_

Round 1
Reviewer 1 Report
Although the subject is interesting, the manuscript has several weaknesses.
The authors attempt to link the role of metabolic hormones with the development of cancer through the effect on macrophage polarization. However, macrophage polarization should be deeply described, from an immunological point of view. The tumor microenvironment (TME) is a very complex scenario where different cells and molecules form a network that is difficult to dissect and simplify to one cell type. The authors should conclude how the different microenvironment generated by these molecules would affect macrophage polarization and then in which way promote or suppress cancer development. The manuscript should include more figures or diagrams. The subject is treated superficially and vaguely; the conclusions summarized in Table 1 are ambiguous. Finally, in the conclusion the authors refer to regulatory effects on macrophages and include "proliferation"; it is important to clarify that macrophages do not proliferate, are non-dividing cells, perhaps the authors refer to a major recruitment of macrophages.
Author Response
Please see the attachment 'Batty et al response to reviewer 1'

Reviewer 2 Report
In their review titled “Metabolic Hormones Modulate Macrophage Inflammatory Responses,” Batty et al review 13 metabolic hormones and discuss their effects on macrophage polarization/regulation of inflammatory response.
This review is useful in that it groups hormones that modulate macrophage pro/anti-inflammatory responses in connection to cancer and obesity. However, the manuscript lacks structure and would greatly benefit from original commentary and/or more elaborate and explicit discussion of the cone connection between these hormones, cancer and obesity.
The authors do not adequately tie together in a seamless way the various themes of the paper. In the opinion of this reviewer some graphics, figures and schemes would greatly elevate this manuscript and make it more cohesive. As it stands, (in the opinion of this reviewer) the authors have summarized a collection of studies but have failed to make their central theme explicit and provide original commentary/opinion that would not be gained by reading the cited studies alone.
Because of this, this reviewer recommends major changes to the paper before publication.
Additionally, there are several distracting grammatical, style, and formatting mistakes. These include (but not limited to):
Several run on sentences, (for instance the sentence beginning with line 80). Consider breaking up into smaller sentences.
Line 61 and 63, define TAM and TME first instance in text
Paragraph starting line 49: Focuses only on macrophages in cancer development but review encompasses cancer and obesity.
Line 89: Reference Li S, 2007 not numbered in line with other references.
Sentence line 96: Line 97, missing closing parenthesis
Line 278, “by”
Line 286 “studied”
Line 319: Insuela 2013 reference not numbered
Line: 599-600 women vs woman
Author Response
Please see the attachment 'Batty et al response to reviewer 2'

Reviewer 3 Report
This review gives an overview of different metabolic hormones affecting macrophages.
It is a very interesting paper and I would just like to make a few remarks.
Some of the factors described lack a direct relationship to the topic of the paper. For example, there is no link established between FGF19 and macrophages; thus, it is not necessary to mention FGF19. In the case of gastrin only a small part of the Gastrin topic addresses macrophages. Another example is glucagon, there is no link to macrophages described.
In addition, to improve the quality of the manuscript and in order to facilitate absorption of knowledge , adding figures would be helpful: To dissect direct and indirect effects: please add figure(s) showing signaling pathways directly regulated by metabolic hormones vs. pathways indirectly regulated. So it should become clear, where direct and strong links between the corresponding hormone and macrophages exist, and where not.
Author Response
Please see the attachment 'Batty et al response to reviewer 3'

Round 2
Reviewer 2 Report
In their revision of the review titled “Metabolic Hormones Modulate Macrophage Inflammatory Responses,” Batty et al have adequately addressed reviewers' comments.
The addition of figures and tables as well as the resutucturng and additional clarifications have made for a more cohesive and informative manuscript.
Some very minor issues include:
Line 54: Separate paragraph title “The role of macrophages in cancer development “
Line 216: New paragraph for Ghrelin
misspelling of "polarization"
Parts of paragraph italicized/bolded
This reviewer recommends the manuscript for publication
Reviewer 3 Report
The manuscript has been significantly improved.